# The role of positively charge poly-L-lysine in the formation of high yield gold nanoplates on the surface for plasmonic sensing application

Marlia Morsin[1,2]*, Suratun Nafisah[3], Rahmat Sanudin[1,2], Nur Liyana Razali[1,2‡], Farhanahani Mahmud[1,2‡], Chin Fhong Soon[1,2‡]

1 Microelectronics & Nanotechnology—Shamsuddin Research Centre (MiNT-SRC), Institute of Integrated Engineering (IIE), Universiti Tun Hussein Onn Malaysia, Parit Raja, Batu Pahat Johor, Malaysia, 2 Faculty of Electronic and Electrical Engineering (FKEE), Universiti Tun Hussein Onn Malaysia, Parit Raja, Batu Pahat Johor, Malaysia, 3 Department of Electrical Engineering, Institut Teknologi Sumatera (ITERA), Lampung Selatan, Indonesia

☯ These authors contributed equally to this work.
‡ These authors also contributed equally to this work.
* marlia@uthm.edu.my

**Data Availability Statement:** All relevant data are within the manuscript and its Supporting Information files.

## Abstract

An anisotropic structure, gold (Au) nanoplates was synthesized using a two-step wet chemical seed mediated growth method (SMGM) directly on the substrate surface. Prior to the synthesis process, poly-l-lysine (PLL) as a cation polymer was used to enhance the yield of grown Au nanoplates. The electrostatic interaction of positive charged by PLL with negative charges from citrate-capped gold nanoseeds contributes to the yield increment. The percentage of PLL was varied from 0% to 10% to study the morphology of Au nanoplates in term of shape, size and surface density. 5% PLL with single layer treatment produce a variety of plate shapes such as hexagonal, flat rod and triangular obtained over the whole substrate surface with the estimated maximum yield up to ca. 48%. The high yield of Au nanoplates exhibit dual plasmonic peaks response that are associated with transverse and longitudinal localized surface plasmon resonance (TSPR and LSPR). Then, the PLL treatment process was repeated twice resulting the increment of Au nanoplates products to ca. 60%. The thin film Au nanoplates was further used as sensing materials in plasmonic sensor for detection of boric acid. The anisotropic Au nanoplates have four sensing parameters being monitored when the medium changes, which are peak position (wavelength shift), intensity of TSPR and LSPR, and the changes on sensing responses. The sensor responses are based on the interaction of light with dielectric properties from surrounding medium. The resonance effect produces by a collection of electron vibration on the Au nanoparticles surface after hit by light are captured as the responses. As a conclusion, it was found that the PLL treatment is capable to promote high yield of Au nanoplates. Moreover, the high yield of the Au nanoplates is an indication as excellent candidate for sensing material in plasmonic sensor.

**Funding:** The research is supported by Ministry of Higher Education (MOHE) Malaysia through Fundamental Research Grant Scheme (FRGS/1/2019/STG07/UTHM/02/1) (MM, FM) (https://mygrants.gov.my) and Hibah Penelitian GBU45 (No. B/515/IT9. C/PT.01.03/2021) (SN) from Institut Teknologi Sumatera. The authors would also like to thank Universiti Tun Hussein Onn Malaysia for partially sponsored this work. The funders had a role in decision to publish the manuscript.

**Competing interests:** The authors have declared that no competing interests exist.

# Introduction

Metal nanoparticles (MNPs) have been employed in many recent and modern applications due to its enthralling potential including photovoltaic [1, 2], sensing [3], catalysis [4], surface-enhanced Raman scattering [5] and biomedical [6]. https://www.synonyms.com/synonym/hithertoNumerous approaches reported in preparing MNPs either top-down (known as physical approach) or bottom-up approach, which referred to chemical approach. The physical method requires high equipment costs and a lengthy preparation process such as lithography [7, 8] while a photolithography process has limited to a predetermined substrate size [9]. The bottom-up is simpler, easier and low cost preparation as compared to physical methods, but need high precision to control chemical reaction such as electrochemical [10], template—directed synthesis [11], photochemical reduction [12] and seed mediated growth method (SMGM) [13–15]. The most widely used chemical method is the SMGM due to its ability to produce high and quality nanoparticles [16].

In the preparation of MNPs using SMGM, there are two main steps namely seeding and growth processes. The reduction agent such as sodium borohydrate and ascorbic acid were commonly used in producing these nanoparticles. Besides, agents like polymer and surfactant; i.e. cetyltrimethyl ammonium bromide (CTAB), hexamethylenetetramine (HMT) and poly (vinyl pyrrolidone) (PVP) were used during the synthesis process to control the growth of NPs and prevent the particles from aggregation. Owing to a tremendous progress in MNPs researches, a variety of structures have been formed using numerous types of metal using SMGM such as gold [17, 18], platinum [19], silver [20] and titanium [21]. Gold nanoparticles (AuNPs) are actively studied because its stability, biocompatibility as well as high carrier capability features. Thus far, the most common synthesized structure of AuNPs is isotropic nanosphericals [22, 23]. The Au nanosphericals are simpler and easier to fabricate as compared to other structures.

Currently, our group have been working with AuNPs specifically in sensing application [24–28]. The basis of this study started with Au nanosphericals and then further expanded by producing anisotropic nanostructures to enhance and expand the scope of applications. The sensing properties is strongly depending on the size and shape in producing their plasmon band [29, 30]. A plasmon is a collective oscillation of free electrons on the metal surface due to the interaction of electrons when hit by the electromagnetic light. Until now, we have successfully synthesized anisotropic Au such as nanorods [24, 25, 31], nanorices [32], nanomesh [33], and nanobipyramids [26, 27, 34]. All of these structures have been produced in different sizes and densities in solution form. However, there are some difficulties to attach them onto substrate as a thin film due to its shapes. Therefore, we have tried to deposit anisotropic gold in the form of thin film for stability purpose and to be used repeatedly similar to isotropic nanosphericals.

Hence, in this research, we employed a cation polymer namely poly l lysine (PLL) prior to the seeding process to produce high yield of Au nanoplates. In the plasmonic sensing application, high density is one of the crucial parameters since it contributes to sensitivity and repeatability as well as stability of the sensor. Previously, many attempts have been done to produce high yield of nanoplates by investigating the seeding process [35], growth aging period [17] and the materials, i.e. surfactants and capping agents used in the synthesis process [28] and also simplifying the process [36]. As a result, we found that the implementation of PLL in the synthesizing process successfully promotes high yield of Au nanoplates product especially for hexagon shapes. Furthermore, the Au nanoplates thin film have been further used in plasmonic sensing application for detection of boric acid.

Boric acid is a dangerous and toxic chemicals used as a pesticide [37] to kill fungi, mites, insects and plants such as cockroaches, wood fungi and flies. Boric acid has been declared unsafe as flavour enhancer in food by the World Food and Health Organization (FAO / WHO) Expert Committee [38]. However, some small scales food industries still used it as preservatives in noodles, meat, seafood and dairy products meat [39]. The lethal dose of boric acid is 3–6 g for infants and 15–20 g for adults [40]. Exposure to boric acid in large quantities for long periods lead to toxic symptoms including vomiting, diarrhoea and abdominal pain. Even though there are other sophisticated boric acids detection method in the like chromatography-mass spectrometry (GC-MS) [41] and high-performance liquid chromatography (HPLC) [42], the direct detection using localized plasmon resonance properties of gold nanoparticles offers faster and easier set up by simply dissolving boric acid powder in water as opposed to the existing detection method.

## Experimental methods

### Materials

The chemicals used for the synthesis are same as published in [17]. The gold source, hydrogen tetrachloroaurate ($HAuCl_4.3H_2O$), cethyltrimethy ammonium boromide (CTAB) poly-L-lysine (PLL) and poly (vinyl pyrrolidone) (PVP) were purchased from Sigma-Aldrich, USA. Trisodium citrate ($C_6H_5Na_3O_7$), sodium tetraborohydride ($NaBH_4$) and ascorbic acid were obtained from Wako Pure Chemical Ltd, Japan. All these chemicals were used as received. The solutions of these chemicals were prepared using deionized (DI) water with resistivity around 18.2 MΩcm which was obtained from pure lab UHQ ELGA water purification system.

### Synthesis of Au nanoplates

In this process, two types of solutions namely seed and growth solution were prepared to form Au nanoplates. The samples were prepared for four different sets of PLL 0.01% (w/v) concentration; 0%, 1%, 5, %, 10% and labelled as PL0, PL1, PL5 and PL10. The percentage of PLL was referred to the ratio used to produce the enhancer solution with DI water. The substrate was immersed in this solution for 30 minutes to impose a positive charge on the substrate surface before proceeding with seeding process. The seeding time was set to 2 hours for all experiments and the growth time was set to 5 hours. The remaining process were identical to the steps reported in our previous study [43].

### Characterization

The gold NPs structure on the substrate was examined by X—ray diffraction (XRD)—D8 Advance Bruker, Germany. The morphology of this sample was analyzed using FESEM—Zeiss Supra 55VP, Germany and the optical characterization was done using Perkin Elmer Lambda 900/UV/VIS/NIR Spectrometer, USA.

### Sensor setup for boric acid detection

All the Au nanoplates thin film samples were used as sensing materials to detect the presence of boric acid. The sensor setup consists of a LS-1 tungsten halogen lamp as a light source, duplex fibre optical probe system, a USB-2000 Ocean Optics spectrometer and a computer with OOIBase32 software as spectrum analyzer tool. In the sensing experiment, the sample was placed in the drawer inside the custom-made sensor chamber. This chamber has two inlets. The first inlet was used to put a fibre optic that connected to the light source. This fibre arm was used to transmit the light source beam towards Au nanoplates sample. The other arm

acted to pass the reflected light to the spectrometer. A hole is made on the top of the chamber to insert the targeted analyte. The sensing sensitivity was based on the change in the optical absorbance of Au nanoplates upon the presence of boric acid in the solution. Boric acid is an odourless white powder that is not flammable or explosive. In this study, the boric acid used was purchased from R&M Chemicals with a purity of 99%. For this sensing application, the boric acid powder is dissolved in DI according to a certain molarity.

## Results and discussion

### Formation of Au nanoplates thin film with variation of PLL concentration

In our trial, we want to introduce the formation of high yield of Au nanoplates with the assistance of polymer cation prior to seeding process. In solution form, PLL contains a positively charged hydrophilic amino group. Thus, unlike a normal growth process of Au nanoplates, the formation of Au nanoplates using PLL promoting effective improvement in the yield especially for bigger size of nanoplates. The phenomena can be understood by the electrostatic interaction of positive charged by poly—l- lysine with negative charges from gold nanoseeds in a citrate ion. As a result, the Au nanoseeds can easily attach onto the positively charged substrate. The schematic of this process is shown in Fig 1.

The structural properties and particles formation of AuNPs on the surface was examined using XRD. The results are shown in Fig 2 and compared to the JCPDS-004-0784 file for bulk gold. An exceedingly high peak at 38.175˚ - 38.125˚ that can be indexed as the (111)—crystallographic planes of face—centered cubic (fcc) gold nanocrystals and another fcc gold crystals planes (200) at 44.225˚ - 44.3˚ were observed. Hence, the products were dominated by most preferred facet; (111) indicating that the plane prefer to oriented in parallel to the surface of the substrate, same as AuNPs formation. Therefore, it can be concluded that a pure crystal of Au nanoplates have been formed on the substrate surface. Table 1 listed all the intensity and angles for all samples.

The results show that the intensities of the two diffraction peaks increased with the increment of PLL up to 5% PLL. The intensity is obtained by observing the highest peaks detected by XRD for each sample and the results have been arranged in sequence for clearer view. When the PLL percentage is increased to 10%, the intensity of diffraction peak was found to decrease. These results prove that the use of PLLs with appropriate percentage help to boost

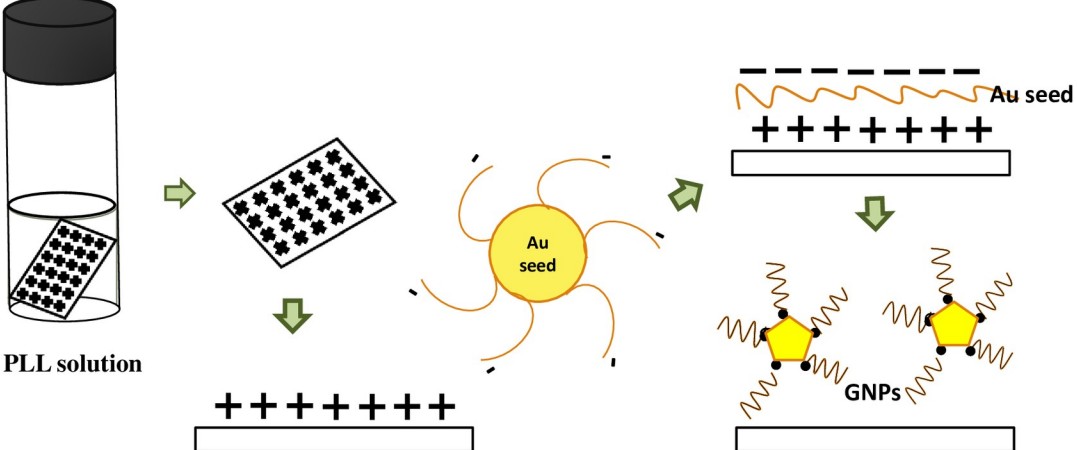

**Fig 1. The schematic process of electrostatic interaction of positive charged by PLL with negative charges from gold nanoseeds in a citrate ion.**

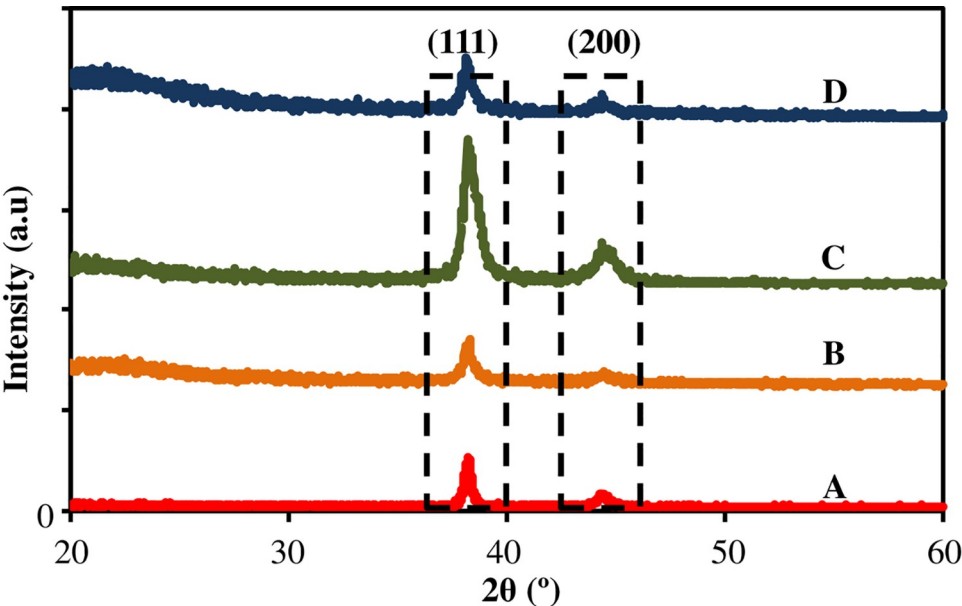

**Fig 2. The XRD of the Au nanoplates grown on the substrates.**

growth of AuNPs on the surface of the substrate. For this study, PL5 is considered as the optimal reading. To support the results of the XRD, morphological characterization was done through FESEM as shown in Fig 3. In these images, we have observed the nanoplates shapes i.e.; hexagonal (including truncated hexagonal) and triangular formation on the substrate. The formation of different plates shapes has been previously mentioned in [43] as the growth of AuNPs started with the formation of triangular shape and grows to a hexagon shape. Other shapes obtained on the surfaces are rod, spherical and irregular shapes and categorized as by-products. Besides, the presence of two groups of Au nanoplates sizes are seen on the growth product. First, the Au nanoplates with edge length more than 150 nm and the other group is the Au nanoplates with smaller edge length (less than 50 nm). The edge length of the hexagonal was varying from 50–200 nm. For the triangular shape, the edge length is 100–350 nm. The thickness (*d*) of the nanoplates is 10–176 nm.

It can be seen that the surface treatment using PLL before seeding process was capable to increase the yield of nanoplates product especially for large size of AuNPs. However, this percentage of nanoplates product was found to be decreased with the high dose of PLL. Besides, it was found that the presence of high concentration PLL causes a tendency to gather seeds from forming nanoplates to spherical or irregular shapes. In this experiment, PLL with 5% has the highest nanoplates yield ~ 47.9% and total AuNPs ~ 76.3%. The analysis for this study was done using ImageJ software to calculate the percentage and surface density. The yield of Au

**Table 1. Intensity and angle for all samples.**

| No. | Sample | Plane (111) | | Plane (200) | |
|-----|--------|-------------|-------------|-------------|-------------|
| | | Angle (Θ°) | Intensity (a.u) | Angle (Θ°) | Intensity (a.u) |
| 1 | PL0 | 38.175° | 99 | 44.275° | 27 |
| 2 | PL1 | 38.175° | 145 | 44.225° | 32 |
| 3 | PL5 | 38.150° | 290 | 44.325° | 84 |
| 4 | PL10 | 38.125° | 120 | 44.300° | 48 |

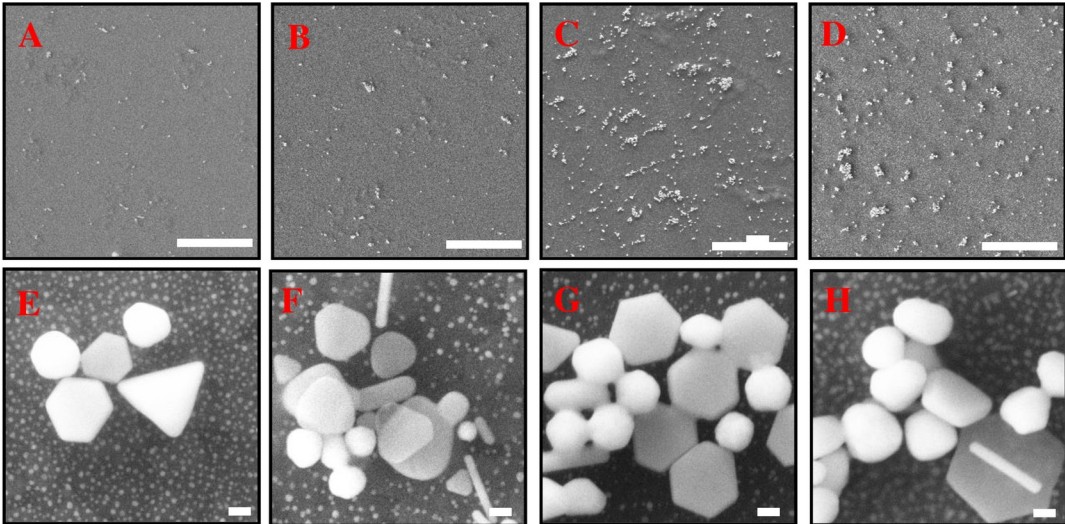

**Fig 3. FESEM images with different concentration of cation polymer, PLL from 0% to 10%.** (A), (E)—PL0, (B), (F)—PL1, (C), (G)—PL5 and (D), (H)—PL10. Scale (A)—(D): 10 μm, (E)—(H)100 nm.

nanoplates and by-products in percentage were plotted in Fig 4. The overall analysis in term of surface density as listed in Table 2.

Finally, the optical responses for all samples have been captured using UV-Vis spectroscopy. The results are shown in Fig 5. It is observed that the optical responses for PL0 and PL5 spectrum have two peaks whereas second peaks for PL1 and PL10 are not very clear. These responses are related with the density of nanoparticles for each sample. The first peak occurs for all Au nanoparticles whereas the second peak is related with the anisotropic nanoparticles. The two peaks assigned as transverse SPR (TSPR) for the first peaks and the second peak is longitudinal SPR (LSPR). The TSPR is free charges vibration in vertical direction of the

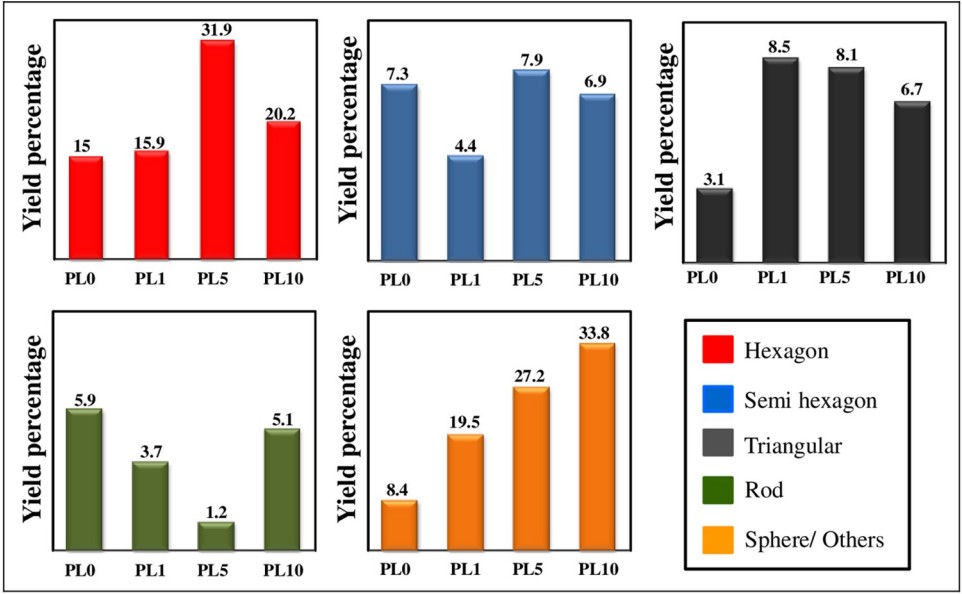

**Fig 4. Analysis for each shape obtained for different concentration of PLL.**

**Table 2. Analysis of surface density on the formation of Au nanoplates with variation of PLL.**

| No | Shape Sample | Hexagon | Semi- hexagon | Traingular | Rod | Sphera / Others | Au Nanoplates / μm² | Total AuNPs / μm² |
|---|---|---|---|---|---|---|---|---|
| 1 | PL0 | 0.53±0.15 | 0.26±0.19 | 0.18±0.09 | 0.28±0.08 | 0.40±0.10 | 0.97±0.26 | 1.58±0.15 |
| 2 | PL1 | 0.56±0.11 | 0.16±0.03 | 0.34±0.23 | 0.13±0.13 | 0.69±0.20 | 1.05±0.18 | 1.87±0.35 |
| 3 | PL5 | 1.12±0.17 | 0.28±0.23 | 0.29±0.13 | 0.04±0.05 | 1.06±0.29 | 1.69±0.32 | 2.68±0.47 |
| 4 | PL10 | 0.71±0.24 | 0.24±0.11 | 0.23±0.04 | 0.18±0.03 | 1.22±0.47 | 1.19±0.77 | 2.60±0.45 |

AuNPs on the surface and LSPR is vibration of free charges in the horizontal direction, i.e., parallel with substrate surface. The interaction between incident electromagnetic wave with nanogold surface has been explained by Chou Chau et al. [44]. The peak resonant wavelengths are determined by elemental composition and aspect ratio by using Drude model [45]. The transverse modes contribute higher intensity with the longitudinal modes dominate the field localization and the transverse modes contribute the field radiation on the surface of the anisotropic nanoparticles.

The direct binding of PLL with Au has not strong electrostatics interactions and it can be improved by assisting the PLL with citrate-capped gold nanoseeds. As a result, the strong electrostatic interactions occurred due to weak acidic condition from citrate [46]. The schematic of positive charge layer with variation concentration of PLL has been deducted in Fig 6. At 0% concentration of PLL, no positive charged layer was formed. When the concentration increased to 1%, the monolayer positive charged was formed on the substrate and when the concentration increased up to 5%, the substrate surface was covered with homogeneous positive charged layer. However, when we increase the concentration to 10%, the positive charges are increased. Hence, the PLLs tend to aggregate each other since the spacing is very narrow causing the non-homogenous positive layer. This situation resulting a phenomenon named

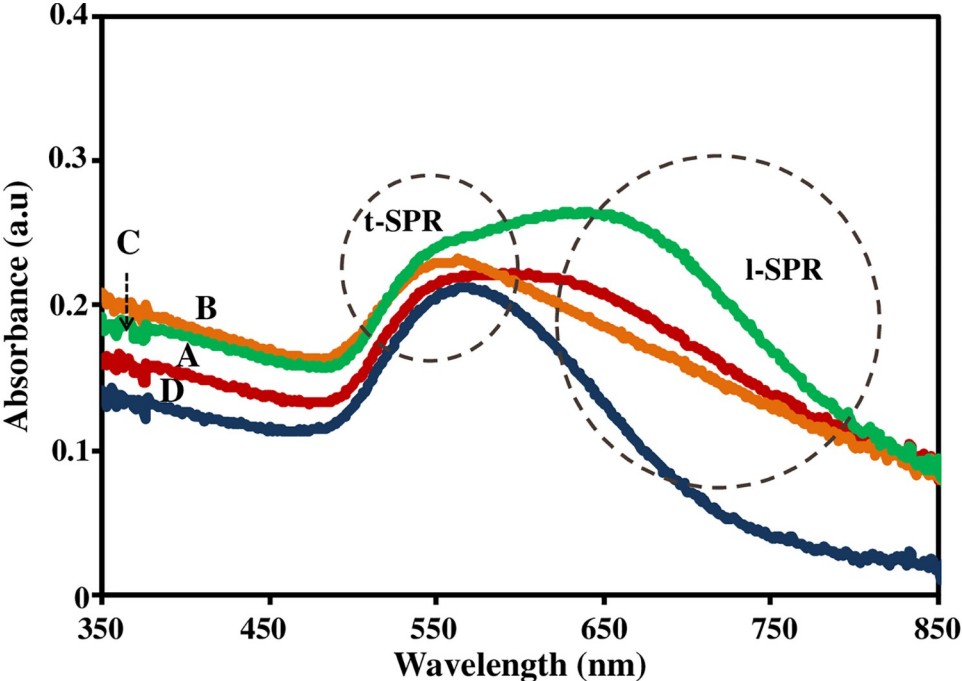

**Fig 5.** The optical response of AuNPs with different PLL concentrations (A) PL0, (B) PL1 (C) PL5 and (D) PL10.

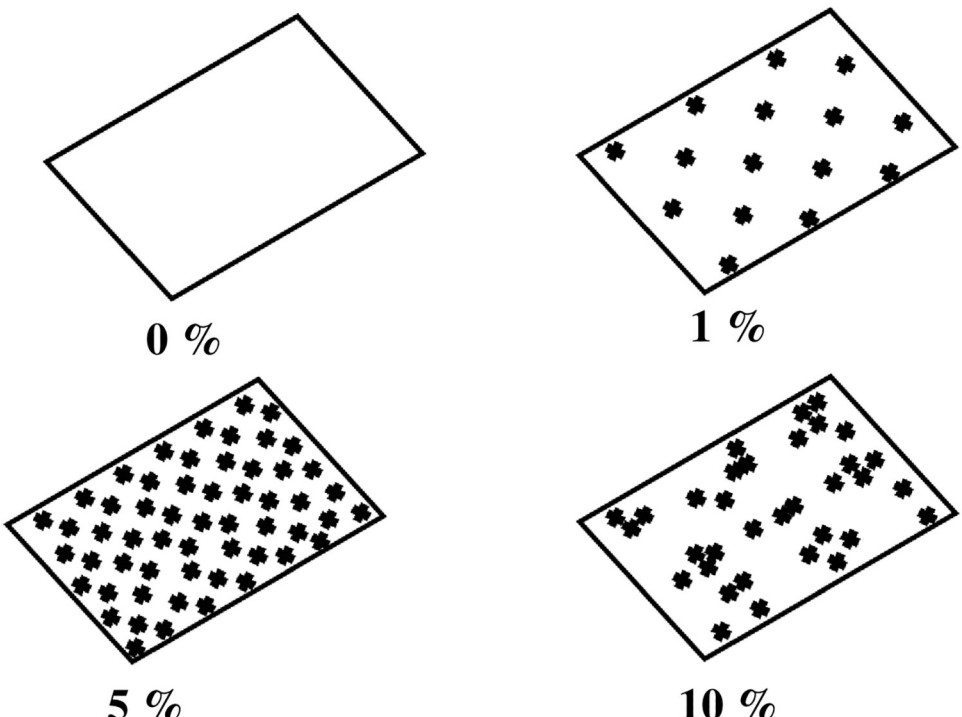

**Fig 6. The schematic of positive charge layer with different percentage of PLL.**

Ostwald ripening [47] that the change of an inhomogeneous gold nanoplates on the surface structure over time.

In order to increase the yield of Au nanoplates, we tried to modify the synthesis method by double the PLL coating process. In this process, the substrate was immersed into PLL solution 5% for 30 minutes same as the previous method and proceed with the seeding process. This process was repeated twice. As a result, it was found the obtained Au nanoplates yield increased to ~ 59.4% and total AuNPs formed on the substrate surface was ~ 93%. It can be seen that the new Au nanoplates grew on the free space on the substrate further increasing the yield. The results are shown in Fig 7.

Next, we tried to triple the PLL layer but the results shown the grown Au nanoplates decreased on the substrate to ~ 52.2% with total NPs ~76.6% (S1 Fig). It was believed that during the third process, the new nanoparticles tend to grow onto the existing nanoparticles resulting stacking nanoparticles. This condition is unacceptable for plasmonic sensing application due to its broad spectrum. Moreover, some of the existing NPS were peeled off during the synthesising process.

### Au nanoplates thin film as a sensing material for boric acid detection

In the sensing applications, the Au nanoplates thin film was used as a sensing material in plasmonic sensor for the detection of boric acid. For this study, the samples used are PL5 single PLL layer and PL5 double PLL layer and labelled as PL5-1 and PL5-2. The selection for these samples because they exhibit dual plasmon resonance with nanoplates percentage more than 40%. The sensor setup used same as reported in our previous work [48] and all the equipment were manually assembled with sensor chamber. The targeted analyte boric acid was prepared in solution form. The chemical reaction of boric acid with air to form a boric acid solution is

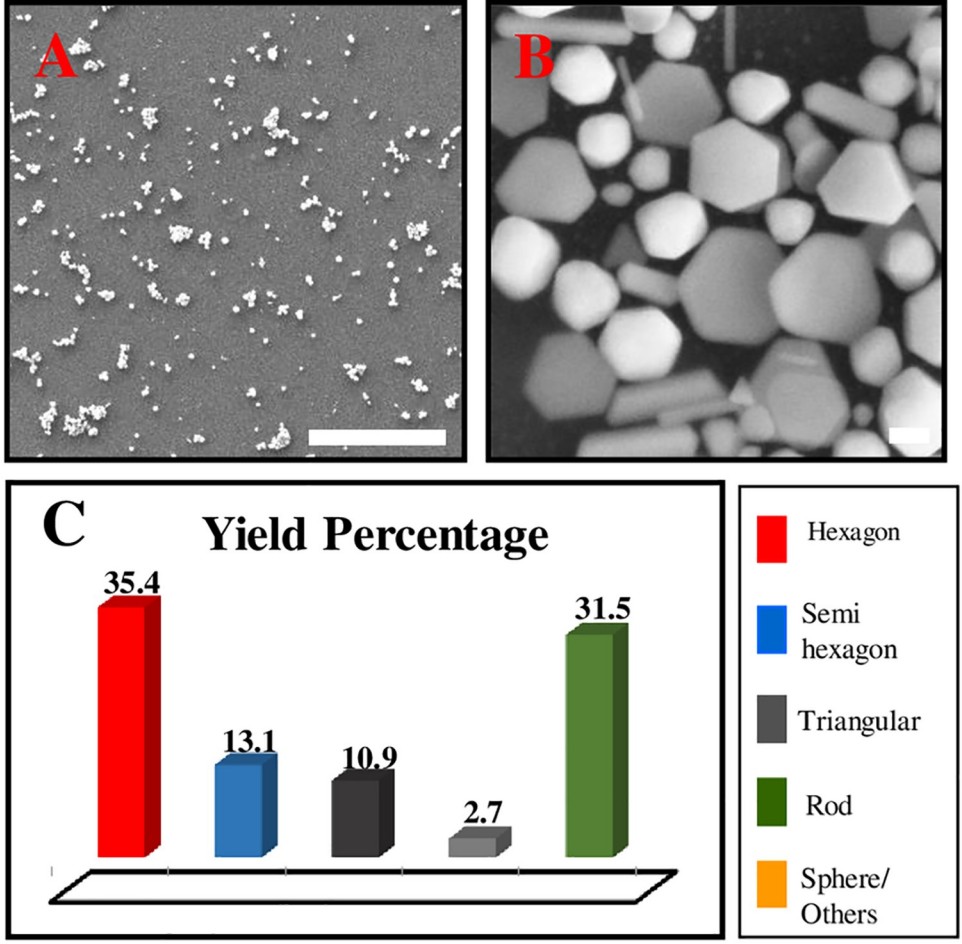

**Fig 7. The FESEM image and yield percentage for the sample double PLL layer.** Scale (A): 10 μm and (B)100 nm.

as follow: $H_3BO_3 + H_2O \rightarrow 3H^+ + BO_3^{3-}$ [49]. The schematic response of the plasmonic sensor based Au nanoplates is shown in Fig 8.

The plasmonic sensor works based on the change of dielectric properties from surrounding medium which is related to refractive index medium. Therefore, in this process, DI water was used as a reference medium and boric acid 10mM (6.184 mg/L) used as a targeted analyte medium. Fig 9 shows the optical sensor responses of PL5-1 and PL5-2 in these two mediums.

From the results, we can see the changes of the optical sensing response in DI water and boric acid medium. The changes are recorded in peak position (wavelength shift) and intensity for both samples. These four parameters are counted as sensing parameters in this sensor response. The changes occurred is depending on the refractive index, $n$. The $n$ for water is 1.333 and 10 mg/L boric acid is 1.3339 with difference is 0.0009. Hence, the wavelength is red-shifted when the medium change from DI water to boric acid as well as the intensity increase for higher $n$. The changes for these sensing responses can be described using Mie theory [27, 28, 43]. Besides, the PL5-2 with higher density of Au nanoplates has good responses towards different medium as compared to PL5-1. The LSPR peak for PL5-1 appeared as slight peak and not clearly seen. Therefore, the yield of Au nanoplates influenced the plasmonic sensor performance towards boric acid.

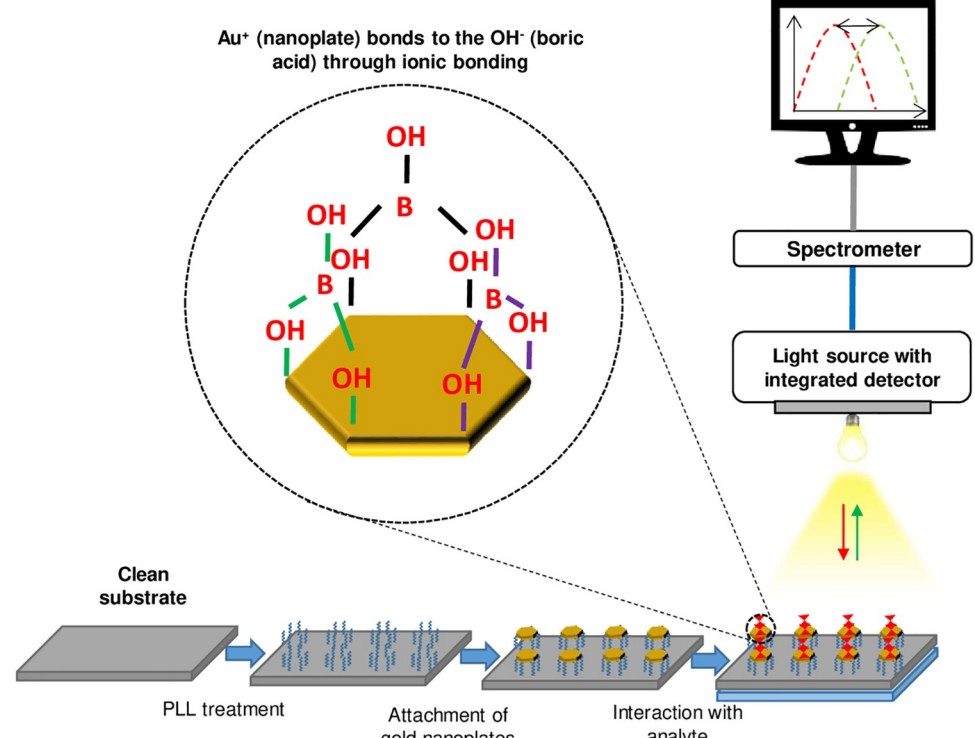

**Fig 8. Schematic of sensing mechanism of plasmonic sensor using Au nanoplates as sensing material for boric acid detection.**

Fig 10 shows the sensing data for the change in the peak position (wavelength shift) and (b) intensity for sample PL5-1 and PL5-2. This data is the mean change of peak position and intensity with DI water as a reference for TSPR and LSPR for 10 samples. In this data, we can clearly see that the change in LSPR peaks are dominant for both samples. As previously mentioned, the LSPR produced by the resonant of free electron in the horizontal direction. The high yield of Au nanoplates contribute to this result. Thus, we found that the intensity is greatly affected by medium changing and wavelength shift is purely depending on the medium refractive index.

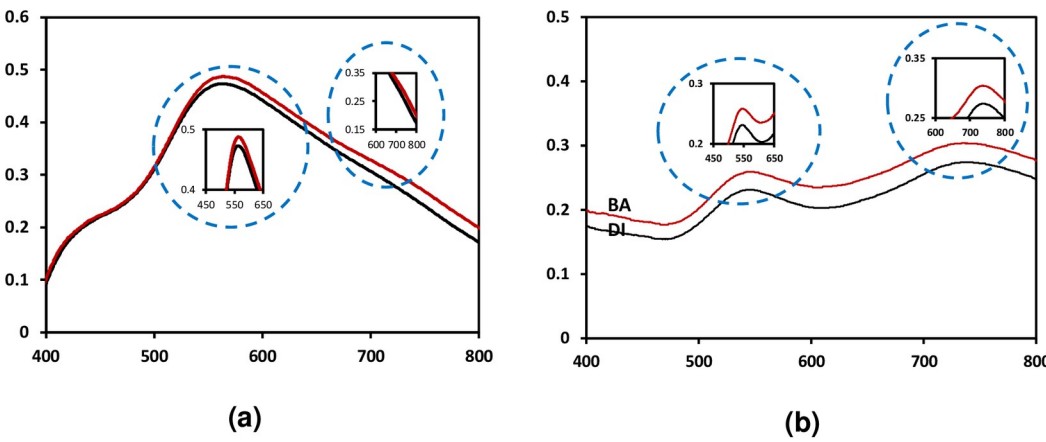

**Fig 9.** The optical sensor responses for (a) PL5-1 and (b) PL5-2 in DI water and boric acid 10 mM.

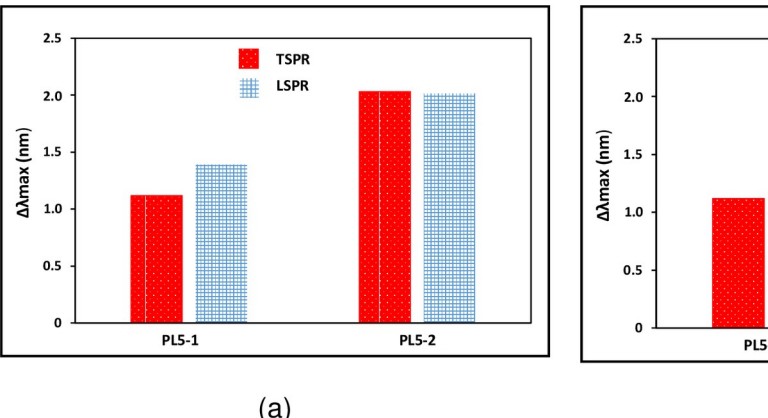
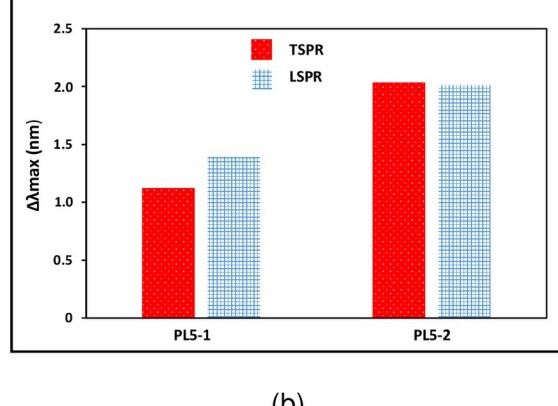

(a)                                                                    (b)

**Fig 10.** Sensing data for the change in (a) peak position (wavelength shift) and (b) intensity for sample PL5-1 and PL5-2.

Then, the PL5-2 was further used for sensitivity test for 5 different concentrations of boric acid as listed in Table 3. The refractive index was calculated using Arago-Biot equation [50].

In this test, the mean change of peak position and intensity with DI water as a reference for TSPR have been plotted in Fig 11(A) and LSPR for Fig 11(B). Both graphs show linear trend from 0.05mM to 50mM where the linear correlation efficient (r) are almost 0.90 but then attenuated at the 150mM. The linear trend is related with the increment of refractive index that affecting the sensing response. For the attenuation at higher concentration, the solution tends to silt in the bottom because it was more concentrated. The upper part of the solution becomes clearer whereas the bottom becomes more concentrated and viscous due to particles agglomeration. Even though the LOD is low as compared to other complicated sensing techniques, the sensing response is fast and in order to enhance the sensitivity, the size and shapes control of gold nanoplates must be improved. Instead of that, based on our previous works, the sensor shows good repeatability response by showing fast response and recovery for at least for five cycles [43]. As compared to isotropic nanoparticles such as nanosphericals that have only transverse localized surface plasmon resonance (TSPR), the anisotropic nanoparticles, for this case nanoplates have additional sensing parameters due to the longitudinal localized surface plasmon resonance (LSPR). These additional parameters enhance the sensitivity of plasmonic sensor thus improve the accuracy and performance of sensing application. Additionally, the anisotropic Au nanoparticles has been intensively used for detection of targeted analyte that showing the potential of their sensing ability [51, 52].

**Table 3. Refractive index for testing medium.**

| No. | Medium | Concentration | | Refractive Index, $n$ |
|---|---|---|---|---|
| | | (mg/L) | (mM) | |
| 1 | DI Water | | | 1.3333 |
| | $\varepsilon_r = 80_{T = 0°C}$ | | | |
| 2 | | 3.092 | 0.05 | 1.333336 |
| 3 | | 30.92 | 0.5 | 1. 333361 |
| 4 | Boric Acid | 309.2 | 5 | 1. 333616 |
| 5 | | 3092 | 50 | 1. 336166 |
| 6 | | 9276 | 150 | 1. 341833 |

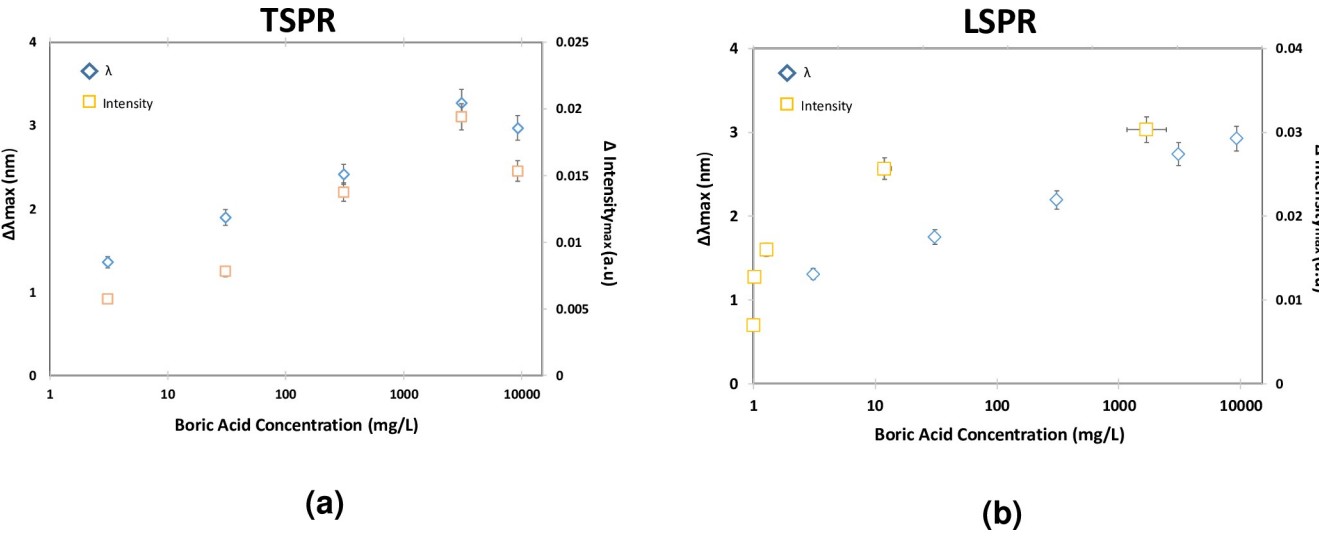

**Fig 11.** The plasmonic responses of (a) TSPR and (b) LSPR with the relationship between peak position and intensity with boric acid concentration.

## Conclusion

In this study, Au nanoplates have been successfully grown on the substrate using seed mediated growth method. As a result, the usage of cation polymer, PLL in seeding process is capable to increase the yield of Au nanoplates especially for large size. Nevertheless, the presence of PLL also stimulates the growth of non nanoplates shapes such as spherical and irregular shapes. Moreover, the repeated PLL treatment increases the yield percentage up to ca. 59%. Therefore, the PLL treatment is able to promote high yield of Au nanoplates. The Au nanoplates thin film is also tested as sensing material in plasmonic sensor for boric acid detection. The sensing responses are monitored by the changes in the peak position (wavelength shift) and intensity of TSPR and LSPR when the medium changes. The anisotropic Au nanoplates has huge potential to be employed in sensing applications due to its plasmonic resonance effect.

## Supporting information

**S1 Fig. The FESEM image and yield percentage for the sample triple PLL layer.** Scale (A): 10 μm and (B) 100 nm.
(DOCX)

## Acknowledgments

The experiment was conducted at Microelectronics & Nanotechnology—Shamsuddin Research Centre (MiNT-SRC), UTHM and Institute of Microengineering and Nanoelectronics (IMEN), Universiti Kebangsaan Malaysia.

## Author Contributions

**Conceptualization:** Marlia Morsin.

**Methodology:** Marlia Morsin.

**Resources:** Nur Liyana Razali, Chin Fhong Soon.

**Visualization:** Marlia Morsin.

**Writing – original draft:** Marlia Morsin, Suratun Nafisah.

**Writing – review & editing:** Rahmat Sanudin, Farhanahani Mahmud.

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
