## [Decision Letter · Decision Letter 0]

25 Sep 2021

PONE-D-21-28832The Role of Positively Charge Poly-L-Lysine in the Formation of High Yield Gold Nanoplates on the Surface for Plasmonic Sensing ApplicationPLOS ONE

Dear Dr. Morsin,

Thank you for submitting your manuscript to PLOS ONE. After careful consideration, we feel that it has merit but does not fully meet PLOS ONE’s publication criteria as it currently stands. Therefore, we invite you to submit a revised version of the manuscript that addresses the points raised during the review process. Please submit your revised manuscript by Nov 09 2021 11:59PM. If you will need more time than this to complete your revisions, please reply to this message or contact the journal office at plosone@plos.org. Please include the following items when submitting your revised manuscript:A rebuttal letter that responds to each point raised by the academic editor and reviewer(s). You should upload this letter as a separate file labeled 'Response to Reviewers'.A marked-up copy of your manuscript that highlights changes made to the original version. You should upload this as a separate file labeled 'Revised Manuscript with Track Changes'.An unmarked version of your revised paper without tracked changes. You should upload this as a separate file labeled 'Manuscript'.

We look forward to receiving your revised manuscript.

Kind regards,

Yuan-Fong Chou Chau

Academic Editor

PLOS ONE

Journal Requirements:

When submitting your revision, we need you to address these additional requirements. 1. Please ensure that your manuscript meets PLOS ONE's style requirements, including those for file naming. The PLOS ONE style templates can be found at https://journals.plos.org/plosone/s/file?id=wjVg/PLOSOne_formatting_sample_main_body.pdf and https://journals.plos.org/plosone/s/file?id=ba62/PLOSOne_formatting_sample_title_authors_affiliations.pdf 2. We suggest you thoroughly copyedit your manuscript for language usage, spelling, and grammar. If you do not know anyone who can help you do this, you may wish to consider employing a professional scientific editing service.  Whilst you may use any professional scientific editing service of your choice, PLOS has partnered with both American Journal Experts (AJE) and Editage to provide discounted services to PLOS authors. Both organizations have experience helping authors meet PLOS guidelines and can provide language editing, translation, manuscript formatting, and figure formatting to ensure your manuscript meets our submission guidelines. To take advantage of our partnership with AJE, visit the AJE website (http://learn.aje.com/plos/) for a 15% discount off AJE services. To take advantage of our partnership with Editage, visit the Editage website (www.editage.com) and enter referral code PLOSEDIT for a 15% discount off Editage services.  If the PLOS editorial team finds any language issues in text that either AJE or Editage has edited, the service provider will re-edit the text for free. Upon resubmission, please provide the following: The name of the colleague or the details of the professional service that edited your manuscript A copy of your manuscript showing your changes by either highlighting them or using track changes (uploaded as a *supporting information* file) A clean copy of the edited manuscript (uploaded as the new *manuscript* file)”  3. We note that the grant information you provided in the ‘Funding Information’ and ‘Financial Disclosure’ sections do not match.  When you resubmit, please ensure that you provide the correct grant numbers for the awards you received for your study in the ‘Funding Information’ section. 4. Thank you for stating the following in the Acknowledgments Section of your manuscript:  "The research is funded by FRGS grant (FRGS/1/2019/STG07/UTHM/02/1) from the Ministry of Education (MoE) Malaysia and Hibah Penelitian GBU45 (No. B/515/IT9. C/PT.01.03/2021) from Institut Teknologi Sumatera. The authors also would like to thank to the Universiti Tun Hussein Onn Malaysia for partially sponsored this work. The experiment was conducted at Microelectronics & Nanotechnology - Shamsuddin Research Centre (MiNT-SRC), UTHM." We note that you have provided funding information that is not currently declared in your Funding Statement. However, funding information should not appear in the Acknowledgments section or other areas of your manuscript. We will only publish funding information present in the Funding Statement section of the online submission form. Please remove any funding-related text from the manuscript and let us know how you would like to update your Funding Statement. Currently, your Funding Statement reads as follows:  "The research is funded by FRGS grant (FRGS/1/2019/STG07/UTHM/02/1) (MM, FM) from the Ministry of Education (MoE) Malaysia (https://mygrants.gov.my) and Hibah Penelitian GBU45 (No. B/515/IT9. C/PT.01.03/2021) from Institut Teknologi Sumatera. The authors also would like to thank to the Universiti Tun Hussein Onn Malaysia for partially sponsored this work. The experiment was conducted at Microelectronics & Nanotechnology - Shamsuddin Research Centre (MiNT-SRC), UTHM" Please include your amended statements within your cover letter; we will change the online submission form on your behalf.

Reviewers' comments:

Reviewer's Responses to Questions

**Comments to the Author**

1. Is the manuscript technically sound, and do the data support the conclusions?

Reviewer #1: Yes

Reviewer #2: Partly

2. Has the statistical analysis been performed appropriately and rigorously? 

Reviewer #1: Yes

Reviewer #2: No

3. Have the authors made all data underlying the findings in their manuscript fully available?

Reviewer #1: Yes

Reviewer #2: Yes

4. Is the manuscript presented in an intelligible fashion and written in standard English?

Reviewer #1: Yes

Reviewer #2: No

5. Review Comments to the Author

Reviewer #1: In this manuscript, the authors proposed an anisotropic structure, gold (Au) nanoplates was synthesized using a two steps wet chemical seed mediated growth method (SMGM) directly on the substrate surface. They claimed that the high yield of Au nanoplates exhibit dual plasmonic peaks response that are associated with transverse and longitudinal localized surface plasmon resonance (TSPR and LSPR). Furthermore, the proposed structure is novel, and the results are valuable and exciting to the readers. However, the necessary references and physical mechanism are absent in the manuscript to explain the related results. In summary, I recommend that the manuscript undergo a major revision to address my comments below before resubmission to this journal.

1. Line 31-33, it is written that “The high yield of Au nanoplates … are associated with transverse and longitudinal localized surface plasmon resonance (tspr and lspr). The “(tspr and lspr)” should be used capital letters (TSPR and LSPR).

2. Line 55-57, it is written that “The physical method requires high equipment costs and a lengthy preparation process such as lithography [7]”. The suitable reference (Nanomaterials (2019),9(12),1691 and Results in Physics (2020) 17, 103116) are suggested to quote regarding this sentence.

3. Line 76-77, it is written that “Currently, our group have been working with AuNPs specifically in sensing application.” The related articles from your group should be quoted with respect to this sentence.

4. Line 79-80, it is written that “The sensing properties is strongly depending on the size and shape in producing their plasmon band.” The suitable references (J. Electromagn. Waves Appl. (2010) 24(8-9) 1005-1014 and Results in Physics (2019) 13, 102290) are suggested to be cited after this sentence.

5. Line 114, it is written that “The chemicals used for the synthesis are same as published in [Procedia]”. What is [Procedia]? Is it a reference? Please check this sentence.

6. Line 124-126, it is written that” The samples were prepared for 6 different sets of PLL 0.01 % (w/v) concentration; 0 %, 1 %, 5, %, 10 % and labelled as PL0, PL1, PL5 and PL10.” Only four different sets are presented in this sentence. Please check it. Besides, it suggests replacing “6” as “six”.

7. Line 175-176, it is written that” Table 1 listed all the intensity and angles for all samples.” Please clarify in more detail that how to measure the intensity.

8. Line 217-218, it is written that” The two peaks assigned as transverse SPR (tspr) for the first peaks and the second peak is longitudinal SPR (lspr). The tspr is free charges vibration in vertical direction of the AuNPs on the surface and lspr is vibration of free charges in the horizontal direction, i.e., parallel with substrate surface.” I think the explanation of the mechanism is not enough. To help the readers to understand the nature of LSPR and TSPR, authors can refer to “J. Appl. Phys. (2016), 120(9), 093110” and “Plasmonics (2008), 3(4), 157-164” or quote them for simplicity.

9. Line 221-230, the authors deduce the surface positive-negative charge pairs corresponding to Fig. 6. The simulation results is suggested to explain this point. If the authors cannot perform the simulations, they can quote the related articles for simplicity (e.g., J. Nanopart. Res. (2018) 20(7), 190 and Nanoscale Research Letters (2016), 11(1),41)

10. The references used in the text should be improved. To be beneficial for the readers to know the other approaches of plasmonic sensors, the suggested articles need to be included in the suitable place of introduction section, i.e., Nanomaterials (2020), 10(3), 493, Results in Physics (2019) 15, 102567, J. Nanopart. Res. (2020) 22(9), 297, Nanomaterials, 2020, 10(7), 1399, J. Phys. D: Appl. Phys. (2021) 54(11) 115301, Nanomaterials, 2020, 10(3), 493 and Results in Physics (2019) 15, 102567. Besides, please recheck the typos throughout the text.

Reviewer #2: This paper describes the design and it also provides some details of a boric acid sensor based on non-spherical gold nanoparticles (plates) attached to a solid surface. The sensitivity of such sensor to boric acid is low and has little practical importance. In contrast to what the authors are stating, the chemistry of nanoparticle stabilization on the surface coated with poly-lysine is more complex than simple electrostatic interaction. This had to be reviewed (this literature is available) and included in the discussion. The reproducibility of measurements are inadequately described in this manuscript. The paper is poorly written, with multiple grammatical errors.

6. PLOS authors have the option to publish the peer review history of their article (what does this mean?). If published, this will include your full peer review and any attached files.

Reviewer #1: No

Reviewer #2: No

---

## [Author Response · Author response to Decision Letter 0]

20 Oct 2021

We are very grateful for the reviews provided by the editor and each of the external reviewers of this manuscript. The comments are encouraging, and the reviewers appear to share our judgement that this study and its results are clinically important. Please see below, our detailed response to comments. All page numbers and line are referred to the manuscript file.

Response to Academic Editors

1. The manuscript have been changed to follow PLOS ONE's style requirements, including those for file naming.

2. The manuscript have been proofread.

3. The funding information as stated below;

The research is supported by Ministry of Higher Education (MOHE) Malaysia through Fundamental Research Grant Scheme (FRGS/1/2019/STG07/UTHM/02/1) (MM, FM) (https://mygrants.gov.my) and Hibah Penelitian GBU45 (No. B/515/IT9. C/PT.01.03/2021) (SN) from Institut Teknologi Sumatera. The authors would also like to thank Universiti Tun Hussein Onn Malaysia for partially sponsored this work. This statement has been added in cover letter for amendment. 

4. The funding statement has been removed in acknowledgement section. 

Response to Reviewers

Reviewer #1

In this manuscript, the authors proposed an anisotropic structure, gold (Au) nanoplates was synthesized using a two steps wet chemical seed mediated growth method (SMGM) directly on the substrate surface. They claimed that the high yield of Au nanoplates exhibit dual plasmonic peaks response that are associated with transverse and longitudinal localized surface plasmon resonance (TSPR and LSPR). Furthermore, the proposed structure is novel, and the results are valuable and exciting to the readers. However, the necessary references and physical mechanism are absent in the manuscript to explain the related results. In summary, I recommend that the manuscript undergo a major revision to address my comments below before resubmission to this journal.

1. The tspr and lspr have been changed to capital letters (TSPR and LSPR).

2.The suggested references (Nanomaterials (2019),9(12),1691 and Results in Physics (2020) 17, 103116) have been cited in the manuscript.

Line 59-60: 

The physical method requires high equipment costs and a lengthy preparation process such as lithography [7-8]

3. Our works with AuNPs specifically in sensing application have been cited in this manuscript.

Line 78-79: 

Currently, our group have been working with AuNPs specifically in sensing application [24-28]. 

4. References; Results in Physics (2019) 13, 102290) and Analyst, 140(2), 386-406 has been included as references and cited in the manuscript.

J. Electromagn. Waves Appl. (2010) 24(8-9) 1005-1014 is not very suitable because the article discuses more on silver and the publication date is more than 5 years.

Line 81-82: 

The sensing properties is strongly depending on the size and shape in producing their plasmon band [29-30].

5.This is technical error and the Procedia have been replaced by ref no.17 

Line 113

17. Procedia engineering, 2017; 184, 637-642. 

6. The number is corrected to four different sets. 

Line 123-124: 

The samples were prepared for four different sets of PLL 0.01 % (w/v) concentration;

7. The reading of the intensity is measured by capturing the highest intensity (a.u) from the data. To prevent misunderstanding, the value of intensity has been removed from the graph because the results are arranged in order to make it clearly view, not starting from the original. The captured data is tabulated in Table 1. The sentence has been added to the paragraph. 

Line 181-183: 

The intensity is obtained by observing the highest peaks detected by XRD for each samples and the results have been arranged in sequence for clearer view.

8. The additional explanation has been added as follow;

Line 225-230: 

The interaction between incident electromagnetic wave with nanogold surface has been explained by Chou Chau et al [44]. The peak resonant wavelengths are determined by elemental composition and aspect ratio using Drude model [45]. The transverse modes contribute higher intensity with the longitudinal modes dominate the field localization and the transverse modes contribute the field radiation on the surface of the anisotropic nanoparticles.

44. Journal of Applied Physics 120, no. 9 (2016): 093110.

45. Kreibig, Uwe, and Michael Vollmer. Optical properties of metal clusters. Vol. 25. Springer Science & Business Media, (2013), p. 100. 

The suggested reference; Plasmonics (2008), 3(4), 157-164 is not very suitable to be cited because the article discuses more on silver and the publication date is more than 5 years.

9. The surface positive-negative charge pairs corresponding to Fig. 6 has been further discussed in the paragraph.

Line 235-245: 

The direct binding of PLL with Au has not strong electrostatics interactions and it can be improved by assisting the PLL with citrate-capped gold nanoseeds. As a result, the strong electrostatic interactions occurred due to weak acidic condition from citrate [46]. The schematic of positive charge layer with variation concentration of PLL has been deducted in Fig 6. At 0 % concentration of PLL, no positive charged layer was formed. When the concentration increased to 1 %, the monolayer positive charged was formed on the substrate and when the concentration increased up to 5 %, the substrate surface was covered with homogeneous positive charged layer. However, when we increase the concentration to 10 %, the positive charges are increased. Hence, the PLLs tend to aggregate each other since the spacing is very narrow causing the non-homogenous positive layer. This situation resulting a phenomenon named Ostwald ripening [47] that the change of an inhomogeneous gold nanoplates on the surface structure over time. 

46. Stobiecka, M. and Hepel, M., 2011. Double-shell gold nanoparticle-based DNA-carriers with poly-L-lysine binding surface. Biomaterials, 32(12), pp.3312-3321.

47. Pattadar, D.K. and Zamborini, F.P., 2019. Effect of size, coverage, and dispersity on the potential-controlled Ostwald ripening of metal nanoparticles. Langmuir, 35(50), pp.16416-16426.

The suggested references are not very suitable because; 

i. J. Nanopart. Res. (2018) 20(7), 190 - focusing on Ag nanoparticles

ii. Nanoscale Research Letters (2016), 11(1),41) - Fe-Doped TiO2 Nanoparticles

and not related with the PLL binding with Au capped citrate as discussed in this manuscript

10. The references for anisotropic gold plasmonic sensor has been added in the manuscript as follow;

Line 334-336:

Additionally, the anisotropic Au nanoparticles has been intensively used for detection of targeted analyte that showing the potential of their sensing ability [51,52]. 

51. Pattanayak, S. and Jana, S.K., 2018. Controllable aqueous synthesis of near-IR-plasmonic anisotropic gold nanoparticles in the hydrazine concentration assisted: hydrazine-citrate hydrogen-bonded network at room temperature and application in highly sensitive SERS-based detection of Pb (II) species. Inorganic and Nano-Metal Chemistry, 48(11), pp.535-540.

52. Sharifi, M., Hosseinali, S.H., Alizadeh, R.H., Hasan, A., Attar, F., Salihi, A., Shekha, M.S., Amen, K.M., Aziz, F.M., Saboury, A.A. and Akhtari, K., 2020. Plasmonic and chiroplasmonic nanobiosensors based on gold nanoparticles. Talanta, 212, p.120782.

The suggested references are more on silver Ag nanoparticles. 

a. Nanomaterials (2020), 10(3), 493, - Ag sensor

b. Results in Physics (2019) 15, 102567, - Ag nanorods

c. J. Nanopart. Res. (2020) 22(9), 297 – Ag squares, 

d. Nanomaterials, 2020, 10(7), 1399- Ag nanorods

e. J. Phys. D: Appl. Phys. (2021) 54(11) 115301 – Ag nanorods

f. Nanomaterials, 2020, 10(3), 493 – Ag nanohair

g. Results in Physics (2019) 15, 102567- Ag veins 

The typing errors have been corrected and this paper has been proofread. 

Reviewer #2

This paper describes the design and it also provides some details of a boric acid sensor based on non-spherical gold nanoparticles (plates) attached to a solid surface. 

1. The explanation has been included in text as follow;

Line 326-328:

 Even though the LOD is low as compared to other complicated sensing techniques, the sensing response is fast and in order to enhance the sensitivity, the size and shapes control of gold nanoplates must be studied. 

2. The electrostatic interaction between PLL and citrate-capped gold nanoseeds has been discussed with suitable references. 

Line 235-245: 

The direct binding of PLL with Au has not strong electrostatics interactions and it can be improved by assisting the PLL with citrate-capped gold nanoseeds. As a result, the strong electrostatic interactions occurred due to weak acidic condition from citrate [46]. The schematic of positive charge layer with variation concentration of PLL has been deducted in Fig 6. At 0 % concentration of PLL, no positive charged layer was formed. When the concentration increased to 1 %, the monolayer positive charged was formed on the substrate and when the concentration increased up to 5 %, the substrate surface was covered with homogeneous positive charged layer. However, when we increase the concentration to 10 %, the positive charges are increased. Hence, the PLLs tend to aggregate each other since the spacing is very narrow causing the non-homogenous positive layer. This situation resulting a phenomenon named Ostwald ripening [47] that the change of an inhomogeneous gold nanoplates on the surface structure over time. 

46. Stobiecka, M. and Hepel, M., 2011. Double-shell gold nanoparticle-based DNA-carriers with poly-L-lysine binding surface. Biomaterials, 32(12), pp.3312-3321.

47. Pattadar, D.K. and Zamborini, F.P., 2019. Effect of size, coverage, and dispersity on the potential-controlled Ostwald ripening of metal nanoparticles. Langmuir, 35(50), pp.16416-16426

3. The reproducibility test has been cited in the text by referring our previous work [43]. 

Line 328-330:

Instead of that, based on our previous works, the sensor shows good repeatability response by showing fast response and recovery for at least for five cycles [43].

43. Sensors, 17(5), p.947.

4. The typing errors have been corrected and this paper has been proofread.

---

## [Decision Letter · Decision Letter 1]

26 Oct 2021

The Role of Positively Charge Poly-L-Lysine in the Formation of High Yield Gold Nanoplates on the Surface for Plasmonic Sensing Application

PONE-D-21-28832R1

Dear Dr. Morsin,

We’re pleased to inform you that your manuscript has been judged scientifically suitable for publication and will be formally accepted for publication once it meets all outstanding technical requirements.

Kind regards,

Yuan-Fong Chou Chau

Academic Editor

PLOS ONE

Additional Editor Comments (optional):

Reviewers' comments:

Reviewer's Responses to Questions

**Comments to the Author**

1. If the authors have adequately addressed your comments raised in a previous round of review and you feel that this manuscript is now acceptable for publication, you may indicate that here to bypass the “Comments to the Author” section, enter your conflict of interest statement in the “Confidential to Editor” section, and submit your "Accept" recommendation.

Reviewer #1: All comments have been addressed

2. Is the manuscript technically sound, and do the data support the conclusions?

Reviewer #1: Yes

3. Has the statistical analysis been performed appropriately and rigorously? 

Reviewer #1: Yes

4. Have the authors made all data underlying the findings in their manuscript fully available?

Reviewer #1: Yes

5. Is the manuscript presented in an intelligible fashion and written in standard English?

Reviewer #1: Yes

6. Review Comments to the Author

Reviewer #1: The authors have revised their manuscript according to my comments. This submission can now be accepted for publication.

7. PLOS authors have the option to publish the peer review history of their article (what does this mean?). If published, this will include your full peer review and any attached files.

Reviewer #1: No

---

## [Editor Report · Acceptance letter]

29 Oct 2021

PONE-D-21-28832R1 

The role of positively charge poly-L-lysine in the formation of high yield gold nanoplates on the surface for plasmonic sensing application 

Dear Dr. Morsin:

I'm pleased to inform you that your manuscript has been deemed suitable for publication in PLOS ONE. Congratulations! Your manuscript is now with our production department. 

Kind regards, 

on behalf of

Dr. Yuan-Fong Chou Chau 

Academic Editor

PLOS ONE